# Deep posterior sampling: Uncertainty quantification for large scale inverse problems

**Jonas Adler**[1,2]                                                        JONASADL@KTH.SE

**Ozan Öktem**[1]                                                          OZAN@KTH.SE

[1] *Department of Mathematics, KTH - Royal institute of Technology, SE-100 44 Stockholm, Sweden*

[2] *Elekta Instrument AB, P.O. Box 7593, SE-103 93 Stockholm, Sweden*

## Abstract

The goal in an inverse problem is to recover a hidden model parameter from noisy indirect observations. Such problems arise in several areas of science and industry and their solutions form the basis for decision making, like when imaging is used in medicine.

Inverse problems are often ill-posed, meaning that there can be multiple solutions consistent with observations and small errors in data result in large errors in the solution. Hence, it is important to assess the uncertainty in the solution of an ill-posed problem and especially so when critical decisions are based on the solution. *Bayesian inversion* offers a coherent framework for both solving an ill-posed inverse problem and quantifying the uncertainty in its solution. Its applicability is however limited by the ability to select a sufficiently 'good' prior and capability to manage the computational burden.

We show how a conditional Wasserstein GAN (WGAN) with a novel minibatch discriminator can be used to sample from the posterior in Bayesian inversion. The suggested approach is demonstrated for image-guided medical diagnostics using computed tomography.

**Keywords:** GAN, Inverse Problem, CT, Image Reconstruction, Bayesian Inversion

## 1. Bayesian Inversion

To formalise the task in an inverse problem, let $x^* \in X$ denote the unknown hidden model parameter we seek and $y \in Y$ is the noisy indirect observations (data). Here, $X$ and $Y$ are appropriate (possibly infinite dimensional) vector spaces whose elements represent possible model parameters and data, respectively. Next, we also have access to a deterministic model $\mathcal{A} \colon X \to Y$ that describes how a model parameter gives rise to data (forward operator) in the absence of noise. Furthermore, assume measured data $y \in Y$ is a single sample of the conditional random variable $(\mathbf{y} \mid \mathbf{x} = x^*)$. Here, $\mathbf{y} = \mathcal{A}(\mathbf{x}) + \mathbf{e}$ where $\mathbf{e} \sim \pi_{\text{noise}}$ denotes a $Y$-valued random variable that represents the noise. Finally, the unknown model parameter $x^* \in X$ itself is assumed to be a sample of an $X$-valued random variable $\mathbf{x} \sim \pi_0$.

The ultimate task in Bayesian inversion is to recover the posterior, e.g. the probability distribution of the conditional random variable $(\mathbf{x} \mid \mathbf{y} = y)$. This formulation comes with several desirable theoretical properties; it is stable with respect to changes in the data even when the inverse problem is ill-posed (Dashti and Stuart, 2017) and the posterior will for many priors concentrate around the true model parameter as noise level goes to zero (consistency & contraction rates) (Nickl, 2017). On the other hand, computations with

the posterior quickly become unfeasible in large scale problems, like those that arise in 3D imaging where images and data are represented by very high dimensional arrays. An option is therefore to explore the (high dimensional) posterior by sampling.

## 2. Deep Posterior Sampling

Most approaches for sampling from a high dimensional posterior are based on Markov chain Monte Carlo (MCMC) techniques. Despite significant theoretical and algorithmic advances (Green et al., 2015; Pereyra et al., 2016), these approaches require access to an explicit prior. Furthermore, they are still not computationally feasible for large-scale inverse problem that arise in 3D imaging.

The deep posterior sampling approach we develop is a conditional variational Bayes method (Blei et al., 2017) where the similarity measure is the Wasserstein 1-distance and the variational family is parametrized by a CNN. The generator is trained against supervised data that are i.i.d. samples $(x_1, y_1), \ldots, (x_m, y_m)$ generated by the random variable $(\mathbf{x}, \mathbf{y}) \sim \mu$. Hence, there is no need to specify the prior as it is implicitly contained in the training data. Stated in more detail, let $\{\mathcal{G}_\theta(y)\}_{\theta \in \Theta}$ be the variational family with parameters $\theta$ that we want to use to approximate the posterior. We emphasize that $\mathcal{G}_\theta(y)$ is not a reconstruction in the traditional sense, rather it is an $X$-valued random variable for each $y$. We want to select the parameters such that the approximation is as close as possible in some distance, where we use the Wasserstein 1-distance, denoted $\mathcal{W}$. We thus (approximately) solve the minimization problem

$$\hat{\theta} \in \underset{\theta \in \Theta}{\arg \min} \, \mathbb{E}_{\mathbf{y} \sim \sigma} \Big[ \mathcal{W}\big(\mathcal{G}_\theta(\mathbf{y}), \pi_{\text{post}}\,(x \mid \mathbf{y})\, dx\big) \Big]. \tag{1}$$

In the above, $\sigma$ is the probability distribution for the $Y$-valued random variable $\mathbf{y}$ that generates data so the above merely states that we seek the best approximation in the variational family for 'all data'. After training, sampling from the posterior $\pi_{\text{post}}\,(x \mid y)\, dx$ can be approximated by sampling from the probability distribution $\mathcal{G}_{\hat{\theta}}(y)$.

Observe now that evaluating the objective in (1) requires access to the very posterior that we seek to approximate. Furthermore, the distribution $\sigma$ of data is often unknown, so an approach based on (1) is essentially useless when the posterior is not explicitly given. Finally, computing the Wasserstein 1-distance directly from its definition is also unfeasible.

*All* of these drawbacks can be circumvented by rewriting (1) as an expectation over the joint law $(\mathbf{x}, \mathbf{y}) \sim \mu$. This makes use of the Kantorovich-Rubenstein duality for the Wasserstein 1-distance and it yields the following approximate version of (1):

$$\hat{\theta} \in \underset{\theta \in \Theta}{\arg \min} \left\{ \sup_{\phi \in \Phi} \mathbb{E}_{(\mathbf{x}, \mathbf{y}) \sim \mu} \Big[ \mathrm{D}_\phi(\mathbf{x}, \mathbf{y}) - \mathbb{E}_{\mathbf{z} \sim \eta} \big[ \mathrm{D}_\phi(\mathrm{G}_\theta(\mathbf{z}, \mathbf{y}), \mathbf{y}) \big] \Big] \right\}. \tag{2}$$

Here, $\mathrm{G}_\theta \colon Z \times Y \to X$ (generator) is a deterministic mapping that parametrises the variational family through a randomised input, i.e., $\mathrm{G}_\theta(\mathbf{z}, y) \sim \mathcal{G}_\theta(y)$ where $\mathbf{z} \sim \eta$ is a $Z$-valued random variable with known distribution. Next, the mapping $\mathrm{D}_\phi \colon X \times Y \to \mathbb{R}$ (discriminator) is a measurable mapping that is 1-Lipschitz in the $X$-variable. Both the generator and discriminator are given as deep neural networks and following (Gulrajani et al., 2017), we

softly enforce the 1-Lipschitz condition on the discriminator by including a gradient penalty term to the training objective in (2). The final step is to replace the $\mu$-expectation with its empirical counterpart that is given by supervised training data. Note however that if the empirical counterpart of (2) is implemented as is, then in practice $\mathbf{z}$ is not used by the generator (so-called mode-collapse). To solve this problem, we introduce a novel discriminator that can be used with conditional WGAN, see (Adler and Öktem, 2018) for the details.

## 3. Clinical image-guided decision making

We apply the method to post-process FBP reconstructions computed from extreme low dose CT data (2 % of normal dose) obtained by sub-sampling and adding noise to Mayo Clinic helical CT data. Samples of the posterior conditioned on a FBP reconstruction are generated using a residual U-Net architecture for the generator and a resnet architecture for the discriminator. We compute the mean and standard deviation from 1000 samples (each 20 ms to generate), followed by a hypothesis test for the presence of an 'anomaly' in the liver. This shows Bayesian inversion is possible for large scale problems.

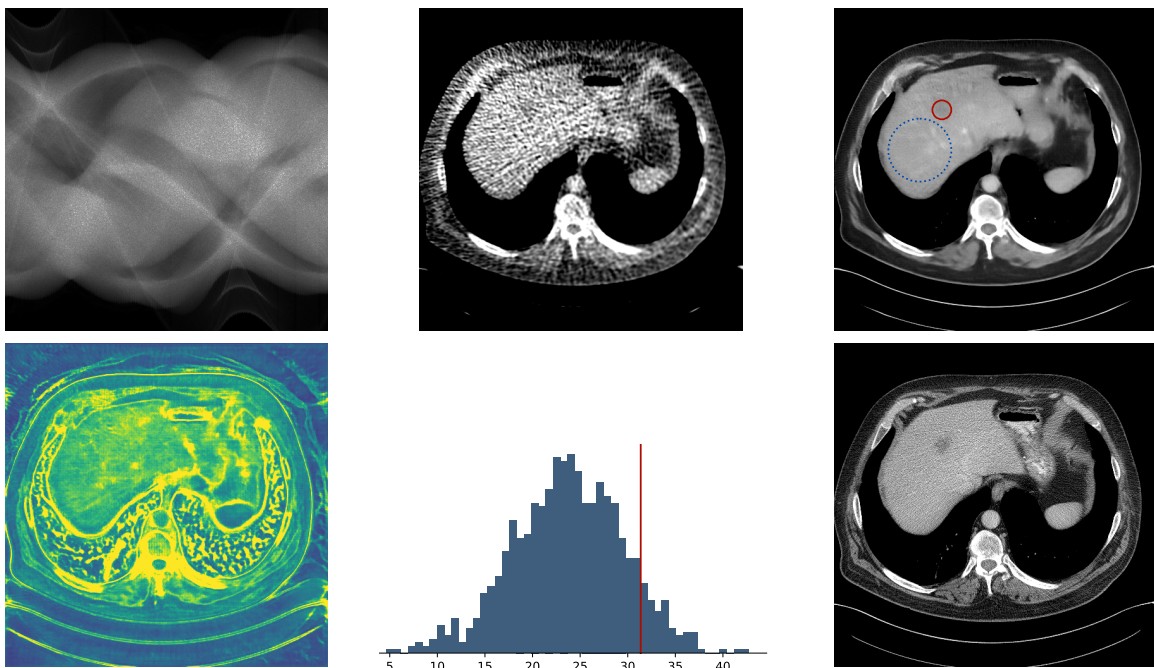

Figure 1: Top row shows a subset of data (left), FBP reconstruction (middle), and sample posterior mean (right) where a suspected anomaly is encircled in red. The bottom row shows the point-wise standard deviation (left). The histogram (middle) shows the distribution of the average HU difference between the region containing the anomaly and a reference region (blue), the red line marks the true difference. A hypothesis test shows that an anomaly with $\Delta$HU $> 10$ exists with $> 95\%$ confidence. The normal dose image (right) confirms the presence of an anomaly.

## Acknowledgments

The work was supported by the Swedish Foundation of Strategic Research grant AM13-0049, Industrial Ph.D. grant ID14-0055 and by Elekta. The authors also thank Dr. Cynthia McCollough, the Mayo Clinic, and the American Association of Physicists in Medicine, and acknowledge funding from grants EB017095 and EB017185 from the National Institute of Biomedical Imaging and Bioengineering, for providing the data.

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
