# OpenReview forum: "Deep posterior sampling: Uncertainty quantification for large scale inverse problems"
_MIDL.io/2019/Conference/Abstract — MIDL Abstract 2019_

### Official Review · AnonReviewer1 · 2019-04-30

**Rating:** 3
**Confidence:** 2

**Review:**

This work applies an inverse problem solution to CT reconstruction in medical imaging. The methodology will be of interest to MIDL audience. However, the demonstration of experimental results is very brief, maybe due to limited space.

---

### Official Review · AnonReviewer2 · 2019-05-01
**Deep posterior sampling applied to low dose CT reconstruction**

**Rating:** 3
**Confidence:** 1

**Review:**

This paper briefly discusses a posterior sampling method based on deep learning. The sampling method uses conditional variational Bayes where the variational family is parameterised by a CNN. The authors apply their method to the problem of low-dose CT reconstruction, showing results on a single image.

Since I'm not an expert in Bayesian inversion, I found the paper a bit difficult to follow and it does not seem to be self-contained (e.g. the new discriminator that can be used with conditional WGAN is not described in the paper; a reference to (Adler and Okten, 2018) is provided instead). This is understandable since this is an abstract paper of 3 pages.

I would have liked to see some aggregated quantitative results about the quality of the reconstruction obtained in a bigger dataset considering more than 1 images.

- Minor comment: There is no definition for the acronym "FBP" (I guess the authors refer to filtered backprojection). This should be defined in the manuscript.

---

### Decision · Program_Chairs · 2019-05-06
**Acceptance Decision**

Accept